

# The association between non-high-density lipoprotein cholesterol to high-density lipoprotein cholesterol ratio (NHHR) and risk of prostate cancer: a retrospective study

Ke Bu[1], Jinru Wang[1], Xiaojie Zheng[2], Kaige Zhang[3], Duolao Wang[4], Hengqing An[3] and Ning Tao[1]

[1] School of Public Health, Xinjiang Medical University, Urumqi, Xinjiang Uighur Autonomous Region, China
[2] Research Management Section, The First Affiliated Hospital of Xinjiang Medical University, Urumqi, Xinjiang Uighur Autonomous Region, China
[3] Urology Department, The First Affiliated Hospital of Xinjiang Medical University, Urumqi, Xinjiang Uighur Autonomous Region, China
[4] Faculty of Medicine, University of Liverpool, Liverpool, Merseyside Metropolitan County, United Kingdom

Corresponding authors
Hengqing An, 9269735@qq.com
Ning Tao, 38518412@qq.com

## ABSTRACT

**Background**. Prostate cancer is one of the most frequent malignancies in the world, with significant morbidity and death rates. Identifying novel biomarkers is critical to reducing morbidity and mortality from the illness today. Although lipids have been linked to an increased risk of prostate cancer, it is unclear if the non-high-density lipoprotein cholesterol (NHDL-C) to high-density lipoprotein cholesterol (HDL-C) ratio (NHHR) is related to prostate cancer. Therefore, we assessed the relationship between NHHR and prostate cancer risk.

**Methods**. This study comprised 1,718 individuals who attended the First Affiliated Hospital of Xinjiang Medical University between March 2020 and March 2024. A pathological examination of a prostate puncture biopsy was utilized to confirm the diagnosis of prostate cancer. The study collected information on participants' clinical and laboratory examinations, used propensity score matching to eliminate potential confounders, and analyzed the relationship between NHHR and prostate cancer, using logistic regression models combined with restricted cubic spline (RCS) and receiver operating characteristic (ROC) curve. Furthermore, sensitivity analyses were undertaken to assess the robustness of the results.

**Results**. (1) There were statistically significant differences in neutrophils, erythrocytes, triglycerides, alkaline phosphatase, and NHHR between the two groups; (2) after adjusting for all covariates, logistic regression revealed a negative association between NHHR and prostate cancer (0.91 (0.83~0.99), $P = 0.028$). Participants in the second quartile had a significantly lower risk of prostate cancer after comprehensive adjustment (0.69 (0.50~0.96), $P = 0.026$); (3) the restrictive cubic spline curve shows a non-linear relationship between NHHR and prostate cancer risk, with a turning point of 1.824; (4) the results of the ROC curve showed that the NHHR had an area under the curve (AUC) of 0.73; the results of the decision curve suggested that the model was able to provide a high benefit value for patients, with a threshold range of approximately 0.01–0.98 and

a maximum net benefit of 0.52, and the calibration curve demonstrated that the model was accurate and reliable.

**Conclusions**. There is a non-linear dose-response relationship between NHHR and prostate cancer risk, which may be associated with a reduced risk of prostate cancer. The finding can be used to detect high-risk groups and prevent prostate cancer.

## INTRODUCTION

Prostate cancer (Pca) is the most common genitourinary malignancy among men, and the second leading cause of cancer death worldwide (*Siegel et al., 2022*). The world age standardized incidence rate is 37.5 per 100,000 population, with higher prevalence rates in regions with high human development indices, such as Europe and North America (*Culp et al., 2020*). The number of new cases of prostate cancer worldwide is expected to increase from 1.4 million in 2020 to 2.9 million globally in 2040 (*James et al., 2024*) . Identifying high-risk individuals and developing simple, affordable, and reliable indicators remain critical to decreasing disease incidence and mortality rates. Advanced age, black ethnicity, and family history are recognized risk factors for prostate cancer (*Gandaglia et al., 2021*).

A literature study revealed that the risk of prostate cancer was dramatically raised in Caucasians over the age of 50, and older adults were more likely to be diagnosed with high-risk prostate cancer (*Perdana et al., 2016*). Data from the National Cancer Registry indicate that black men have 60–70% and 100–120% higher prostate cancer incidence and cancer specific mortality, respectively, compared with Caucasians in the United States (*Hammarlund et al., 2024*; *Yang et al., 2024*). In addition, cardiovascular risk factors are also associated with the risk of prostate cancer, including body mass index (BMI) obesity and triglycerides, high-density lipoprotein cholesterol (HDL-C), low-density lipoprotein cholesterol (LDL-C), *etc*. A meta-analysis including 13 studies showed that high BMI and obesity can increase prostate cancer risk (*Ramadani, Perdana & Ringoringo, 2024*). Similarly, epidemiological evidence suggests that there is polygenic overlap between blood lipids and prostate cancer (*Andreassen et al., 2014*), it means that blood lipids are greatly involved in the occurrence and development of prostate cancer. However, there are differing views on the favorable and negative hazards of blood lipids with prostate cancer. In a meta-analysis of nine studies, three studies discovered a significant positive correlation between LDL-C and the incidence of prostate cancer, while five studies found no correlation between HDL-C and the incidence of prostate cancer (*Kotani et al., 2013*). Therefore, developing more reliable lipid parameters as the predictor of prostate cancer, and exploiting its potential utility in prostate cancer risk assessment could benefit a large number of prostate cancer patients. Non-high-sensity lipoprotein cholesterol (NHDL-C) to high-density lipoprotein cholesterol (HDL-C) ration (NHHR) is a new and developing composite index of atherogenic lipids, it was derived from the ratio of non-high-density

lipoprotein cholesterol (NHDL-C) to high-density lipoprotein cholesterol (HDL-C) (*Kotani et al., 2013*), and NHDL-C is calculated by removing HDL-C from the total cholesterol. Compared with standard lipid parameters, NHHR has demonstrated good predictive and diagnostic efficacy in assessing the risk of atherosclerosis, non-alcoholic fatty liver disease, insulin resistance and metabolic syndrome (*Kim et al., 2013*; *Sheng et al., 2022*). However, there is currently no research reporting the relationship between NHHR and prostate cancer risk. We speculate that there may be a correlation between NHHR and prostate cancer, and NHHR may become an effective detection indicator for prostate cancer prevention and early identification, which is crucial for strengthening current prevention and treatment strategies for prostate cancer. Therefore, based on the retrospective study, we explored the correlation between NHHR and prostate cancer.

## MATERIAL AND METHODS

### Study design and population

This study has been approved by the Ethics Committee of the First Affiliated Hospital of Xinjiang Medical University (20220308-166). Medical records were obtained retrospectively from male patients with complete medical records who visited the First Affiliated Hospital of Xinjiang Medical University between March 2020 and March 2024. All participants underwent prostate biopsy puncture at our hospital. Based on histopathological and immunohistochemical diagnosis, we selected patients with benign prostatic hyperplasia (BPH) as the control group, and prostate cancer patients as the case group. All participants had their blood drawn on an empty stomach the next morning after admission for a variety of physiological and biochemical tests, including the levels of serum total cholesterol, LDL-C, HDL-C, and triglyceride, measured by the automatic biochemical analyzer. The study eventually comprised 1,718 patients, 753 of whom had prostate cancer and 965 who were healthy controls. To keep the baseline information of the case and control groups equal, we used propensity score one-to-one matching, yielding 746 people in each group, all participants have signed the informed consent forms.

### Covariates

The NHHR is calculated as the ratio of non-high-density lipoprotein to high-density lipoprotein, where non-HDL-C is determined by the numerical difference between triglyceride TC (mmol/L) and high-density lipoprotein HDL-C (mmol/L). We have collected basic information of the participants, including age, race, hypertension, diabetes, coronary artery disease and tumor history. In addition, we obtained biochemical markers associated with prostate cancer, including neutrophil count, lymphocyte count, erythrocyte count, serum albumin, globulin, alanine aminotransferase (ALT), aspartate aminotransferase (AST), alkaline phosphatase (ALP), and initial serum prostate-specific antigen (PSA). Multiple imputation is used to replace missing values for variables with a missing rate of less than 20%, and variables with a missing rate exceeding 20% are deleted.

### Propensity score matching analysis

To eliminate confounders, propensity score matching (PSM) was applied to demographic characteristics, these variables included age, ethnicity, hypertension, diabetes mellitus

(DM), coronary artery disease (CAD), and tumor history. Prostate cancer patients and healthy controls are matched 1:1 using nearest neighbor matching, the caliper value is 0.25. Each group had 746 individuals after matching. Standardized mean deviation (SMD) was used to estimate the balance of control variables before and after matching. Typically, a maximum SMD around 0.1 was considered feasible.

## Statistical analysis

Participant characteristics were considered according to the presence or absence of prostate cancer. Normal continuous variables were expressed as mean ± standard deviation (SD), and differences between the two groups were compared using the $t$-test or Mann–Whitney U rank sum test. Categorical variables were expressed as numbers (percentages) and differences between the two groups were compared using the chi-square test. To estimate the association between NHHR and prostate cancer, NHHR was divided into four quartiles (Q1 (<2.15 mmol/L), Q2 (2.15–2.93 mmol/L), Q3 (2.98–3.85 mmol/L), and Q4 (>3.85 mmol/L)). Logistic regression analysis was used to evaluate independent correlations between NHHR and prostate cancer. All variables were first subjected to univariate logistic regression analysis. Multivariate logistic regression was then used to examine factors that were independently linked to the disease; variables that scored $P < 0.05$ in univariate analyses were included in multifactorial logistic regression. Then the model was adjusted according to neutrophils, lymphocytes, erythrocytes, triglycerides, albumin, globulin, AST, ALT, and ALP. According to the calculation of variance inflation factor (VIF), the existence of multicollinearity in logistic regression is diagnosed. Modified Poisson regression is used to re-analyze in sensitivity analyses. We plotted the restricted cubic spline (RCS) to investigate the nonlinear association between prostate cancer risk and NHHR. Next, on either side of these turning points, we created segmented logistic regression models. Lastly, we used receiver operating characteristic (ROC) curves, decision curve analysis (DCA) curves, and calibration curves to thoroughly assess the clinical usefulness and diagnostic performance of NHHR for prostate cancer prediction.

The data were analyzed using SPSS 29.0 and R 4.0.5 software. 'RMS' package was used to plot RCS curves and investigate threshold effects, 'pROC' and 'ggplot2' packages were used to plot ROC curve.

# RESULTS

## Baseline characteristics

A total of 1,718 participants were included in this study, 56.2% (965) of the normal group and 43.8% (753) of the prostate cancer group, and the difference in age between the two groups was statistically significant, with a mean age of 75.41 ± 9.55 years for the normal group and 78.36 ± 9.02 years for the prostate cancer group. After propensity score matching, there were 746 individuals in each group, and the result showed that there were more comorbidities in the prostate cancer group than in the normal group, but there was no significant difference. The prostate cancer group had lower NHHR and higher HDL-C than the normal group (Table 1).

**Table 1** Baseline characteristics stratified by the groups with and without prostate cancer.

| Variable | Unmatched population | | | | Matched population | | | |
|---|---|---|---|---|---|---|---|---|
| | Nomal (*n* = 965) | Pca (*n* = 753) | *P* | SMD | Nomal (*n* = 746) | Pca (*n* = 746) | *P* | SMD |
| Age | 75.41 ± 9.55 | 78.36 ± 9.02 | <.001 | 0.327 | 78.03 ± 8.35 | 78.29 ± 9.02 | 0.557 | 0.029 |
| Neutrophils | 3.98 (3.17, 5.17) | 3.76 (3.04, 4.78) | <.001 | −0.177 | 4.03 (3.22, 5.21) | 3.75 (3.04, 4.77) | <.001 | −0.215 |
| Lymphocyte | 1.64 (1.29, 2.06) | 1.62 (1.22, 2.00) | 0.154 | −0.063 | 1.59 (1.23, 2.02) | 1.62 (1.21, 1.99) | 0.902 | −0.019 |
| Erythrocyte | 4.64 (4.28, 4.94) | 4.47 (4.07, 4.84) | <0.001 | −0.267 | 4.59 (4.22, 4.90) | 4.47 (4.07, 4.84) | <.001 | −0.199 |
| Cholesterol | 4.11 ± 0.98 | 4.02 ± 1.20 | 0.111 | −0.072 | 4.04 ± 0.98 | 4.02 ± 1.20 | 0.605 | −0.021 |
| Triglyceride | 1.29 (0.95, 1.80) | 1.40 (1.03, 2.01) | 0.001 | 0.146 | 1.23 (0.94, 1.75) | 1.40 (1.03, 2.01) | <.001 | 0.236 |
| LDL-C | 2.65 (2.09, 3.19) | 2.71 (2.14, 3.34) | 0.034 | 0.171 | 2.61 (2.09, 3.11) | 2.71 (2.14, 3.34) | 0.004 | 0.175 |
| HDL-C | 1.02 (0.84, 1.22) | 1.05 (0.86, 1.30) | 0.004 | 0.218 | 1.02 (0.85, 1.22) | 1.05 (0.85, 1.30) | 0.022 | 0.210 |
| Albumin | 67.89 (63.30,72.45) | 68.10 (63.03, 72.90) | 0.926 | −0.098 | 67.30 (62.89,71.80) | 68.15 (63.05, 72.97) | 0.099 | −0.052 |
| Globulin | 39.30 (35.80,42.10) | 38.90 (35.10, 42.15) | 0.143 | −0.185 | 38.80 (35.10, 41.68) | 38.90 (35.10, 42.16) | 0.781 | −0.143 |
| AST | 20.50 (16.70,25.90) | 22.60 (17.60, 29.61) | <0.001 | 0.196 | 20.30 (16.40, 25.51) | 22.60 (17.60, 29.58) | <.001 | 0.208 |
| ALT | 19.30 (13.87,28.40) | 19.24 (13.80, 28.85) | 0.744 | 0.112 | 18.67 (13.40,27.8) | 19.29 (13.80,28.89) | 0.202 | 0.141 |
| ALP | 73.32 (59.65, 90.20) | 77.56 (62.00, 103.00) | <0.001 | 0.230 | 74.51 (61.00, 90.81) | 77.58 (62.08, 103.00) | <.001 | 0.221 |
| PSA | 9.71 (5.56, 15.11) | 15.30 (5.34,61.27) | <0.001 | 0.250 | 10.08 (5.75, 15.53) | 15.23 (5.30, 60.91) | <.001 | 0.249 |
| NHHR | 3.19 ± 1.38 | 2.95 ± 1.51 | <0.001 | −0.159 | 3.11 ± 1.35 | 2.95 ± 1.52 | 0.036 | −0.103 |
| Nation | | | 0.775 | | | | 0.583 | |
| No | 646 (66.94) | 509 (67.60) | | 0.014 | 492 (65.95) | 502 (67.29) | | 0.029 |
| Yes | 319 (33.06) | 244 (32.40) | | −0.014 | 254 (34.05) | 244 (32.71) | | −0.029 |
| Hypertension | | | 0.185 | | | | 0.823 | |
| No | 691 (71.61) | 517 (68.66) | | −0.064 | 516 (69.17) | 512 (68.63) | | −0.012 |
| Yes | 274 (28.39) | 236 (31.34) | | 0.064 | 230 (30.83) | 234 (31.37) | | 0.012 |
| Diabetes | | | <0.001 | | | | 0.133 | |
| No | 854 (88.50) | 623 (82.74) | | −0.152 | 641 (85.92) | 620 (83.11) | | −0.075 |
| Yes | 111 (11.50) | 130 (17.26) | | 0.152 | 105 (14.08) | 126 (16.89) | | 0.075 |
| Coronary heart disease | | | 0.003 | | | | 0.130 | |
| No | 901 (93.37) | 673 (89.38) | | −0.130 | 685 (91.82) | 668 (89.54) | | −0.074 |
| Yes | 64 (6.63) | 80 (10.62) | | 0.130 | 61 (8.18) | 78 (10.46) | | 0.074 |
| Tumor history | | | 0.109 | | | | 0.346 | |
| No | 927 (96.06) | 711 (94.42) | | −0.071 | 712 (95.44) | 704 (94.37) | | −0.047 |
| Yes | 38 (3.94) | 42 (5.58) | | 0.071 | 34 (4.56) | 42 (5.63) | | 0.047 |

### Univariate and multivariable logistic regression analysis

The regression model's hypothesis test reveals that the independent variables are not collinear, and it meets the logistic regression requirements. Prior to matching, univariate logistic regression analysis revealed a strong negative correlation between NHHR levels and prostate cancer risk (OR 0.89, 95% CI [0.83–0.95], $P < 0.001$). Other important parameters included neutrophil count, red blood cell count, triglycerides, low-density lipoprotein, albumin, globulin, AST, ALT and alkaline phosphatase. In the multivariate logistic regression, the correlation between NHHR and prostate cancer risk was still statistically significant (OR: 0.87, 95% CI [0.80–0.95], $P = 0.002$) (Table 2), and the same trend was also observed in the data after matching (OR: 0.91, 95% CI [0.83–0.99], $P = 0.028$). When NHHR was divided into quartiles, participants in the second quartile faced greater risk, with a 33% reduction in prostate cancer risk compared to participants in the lowest quartile (OR: 0.67, 95% CI [0.50–0.90], $P = 0.007$); after full adjustment after modeling, this association remained (OR: 0.69, 95% CI [0.50–0.96], $P = 0.026$) (Table 3).

### Sensitivity analysis

We did multiple sensitivity studies to ensure that the results were reliable. To prevent confounding effects, we first used propensity score matching for age, diabetes mellitus, hypertension, coronary heart disease, and history of other neoplasms. Subsequently, we performed univariate and multivariate logistic regression of NHHR by quartiles, in unadjusted model (OR: 0.67, 95% CI [0.50–0.90], $P = 0.007$); after adjusting for all variables, the results had the same trend, but differed only in the second quartile (OR: 0.69, 95% CI [0.50–0.96], $P = 0.026$) (Table 3). Finally, the modified Poisson regression that we performed shows that our results were stable (Table S1). This demonstrates that NHHR is an independent risk factor for prostate cancer.

### Non-linear relationship between NHHR and prostate cancer

Prostate cancer risk and NHHR displayed a nonlinear connection (nonlinear $P < 0.001$), according to the results of the restricted cubic spline regression of NHHR (Fig. 1). The risk of prostate cancer decreased as the NHHR level rose. The change rate of prostate cancer risk dramatically dropped when the NHHR was higher than 1.824. The results of the two-stage logistic regression model showed that when NHHR < 1.824, there was a correlation between NHHR and prostate cancer ($P < 0.001$, OR: 0.48, 95% CI [0.353, 0.632]). When NHHR > 1.824, the correlation between NHHR and prostate cancer was not statistically significant ($P = 0.174$, OR:1.063, 95% CI [0.974–1.162]) (Table 4).

### Predictive value and clinical utility analysis of NHHR

We constructed ROC curves to determine the predictive value of NHHR for prostate cancer. The results show that NHHR has a high predictive value (AUC, 0.73; 95% CI [0.70–0.76]) (Fig. 2A), and the sensitivity of the ROC curve is 86%, indicating that NHHR has good value in detecting prostate cancer patients (Table S2). Furthermore, we computed the model's decision curve and calibration curve, and the findings indicated that the model might offer patients a high benefit value; the threshold range is roughly 0.01–0.98, and the

**Table 2 Univariate and multivariable logistic regression analysis of baseline variables and risk of prostate cancer in the matched and unmatched populations.**

| Variables | Unmatched population | | | | Matched population | | | |
|---|---|---|---|---|---|---|---|---|
| | Univariate | | Multivariate | | Univariate | | Multivariate | |
| | OR (95% CI) | *P* | OR (95% CI) | *P* | OR (95% CI) | *P* | OR (95% CI) | *P* |
| Age | 1.03 (1.02~1.05) | <0.001 | 1.03 (1.02~1.04) | <0.001 | 1.00 (0.99~1.02) | 0.557 | | |
| Neutrophils | 0.93 (0.88~0.97) | 0.002 | 0.86 (0.81~0.91) | <0.001 | 0.91 (0.86~0.96) | <0.001 | 0.86 (0.81~0.92) | <0.001 |
| Lymphocyte | 0.91 (0.79~1.05) | 0.187 | | | 0.97 (0.83~1.13) | 0.728 | | |
| Erythrocyte | 0.60 (0.51~0.71) | <0.001 | 0.76 (0.62~0.93) | 0.009 | 0.70 (0.58~0.83) | <0.001 | 0.76 (0.61~0.94) | 0.010 |
| Triglyceride | 1.16 (1.05~1.27) | 0.002 | 1.34 (1.19~1.51) | <0.001 | 1.33 (1.18~1.48) | <0.001 | 1.45 (1.26~1.66) | <0.001 |
| Albumin | 0.99 (0.98~0.99) | 0.022 | 1.01 (1.01~1.02) | 0.025 | 1.00 (0.99~1.00) | 0.261 | | |
| Globulin | 0.97 (0.96~0.99) | <0.001 | 1.01 (0.99~1.02) | 0.738 | 0.98 (0.97~0.99) | 0.002 | 1.01 (1.00~1.03) | 0.106 |
| AST | 1.02 (1.01~1.02) | <0.001 | 1.01 (1.01~1.02) | 0.002 | 1.02 (1.01~1.03) | <0.001 | 1.02 (1.01~1.03) | 0.003 |
| ALT | 1.01 (1.01~1.01) | 0.011 | 1.00 (0.99~1.00) | 0.541 | 1.01 (1.01~1.01) | 0.003 | 1.00 (0.99~1.00) | 0.444 |
| ALP | 1.01 (1.01~1.01) | <0.001 | 1.01 (1.01~1.01) | 0.004 | 1.01 (1.01~1.01) | <0.001 | 1.01 (1.01~1.01) | 0.026 |
| PSA | 1.02 (1.02~1.03) | <0.001 | 1.02 (1.02~1.03) | <0.001 | 1.02 (1.02~1.03) | <0.001 | 1.02 (1.02~1.03) | <0.001 |
| NHHR | 0.89 (0.83~0.95) | <0.001 | 0.87 (0.80~0.95) | 0.002 | 0.93 (0.86~0.99) | 0.037 | 0.91 (0.83~0.99) | 0.028 |

**Table 3  The associations between NHHR quartile and prostate cancer risk.**

| Variables | Unadjusted OR (95% CI) | P value | Model 1 OR (95% CI) | P value |
|---|---|---|---|---|
| NHHR (Quartile) | | | | |
| Q1 | Ref | | Ref | |
| Q2 | 0.67 (0.50~0.90) | 0.007 | 0.69 (0.50~0.96) | 0.026 |
| Q3 | 0.83 (0.62~1.10) | 0.199 | 0.77 (0.55~1.06) | 0.110 |
| Q4 | 0.94 (0.71~1.26) | 0.685 | 0.82 (0.58~1.15) | 0.246 |

Notes.

OR, Odds Ratio; CI, Confidence Interval.

Model 1: Adjust: Leukocyte, Neutrophils, Erythrocyte, Triglyceride, Albumin, Globulin, AST, ALT, ALP, PSA.

**Table 4  The result of the two-piecewise linear regression model.**

| Outcome: prostate cancer | OR (95% CI) | P-value |
|---|---|---|
| Fitting model by standard linear regression | 0.927 (0.863, 0.995) | 0.037 |
| Inflection point of NHHR | 1.824 | |
| < | 0.48 (0.353, 0.632) | <0.001 |
| ≥ | 1.063 (0.974,1.162) | 0.174 |
| P for log-likelihood ration test | <0.001 | |

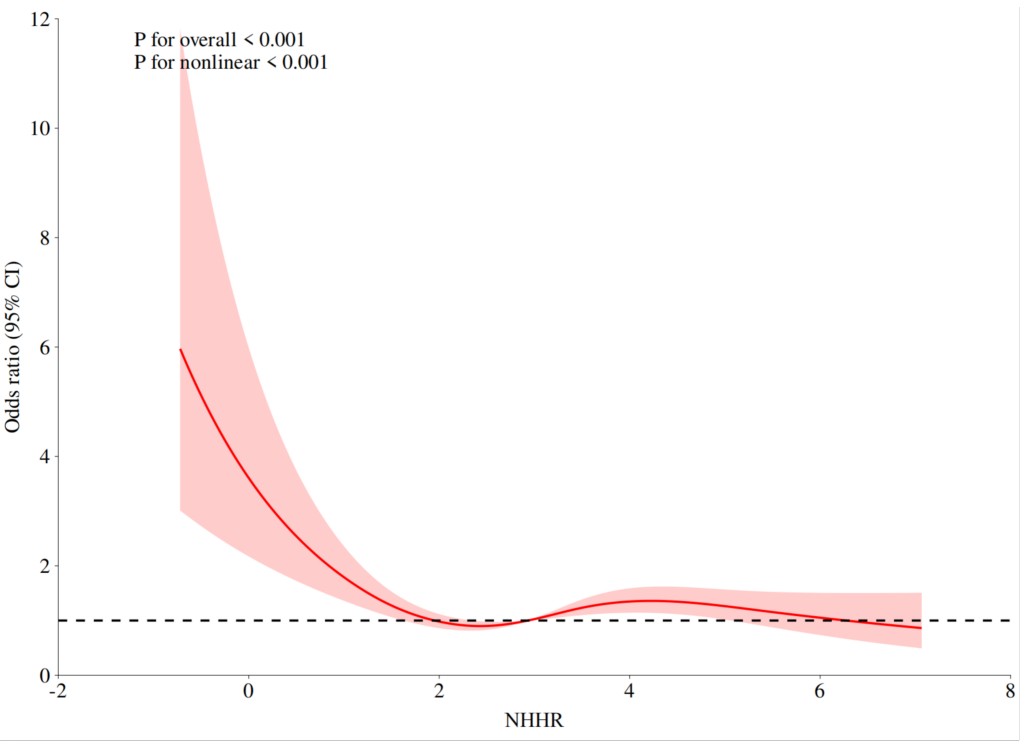

**Figure 1  The association between NHHR and prostate cancer odds ratio.**

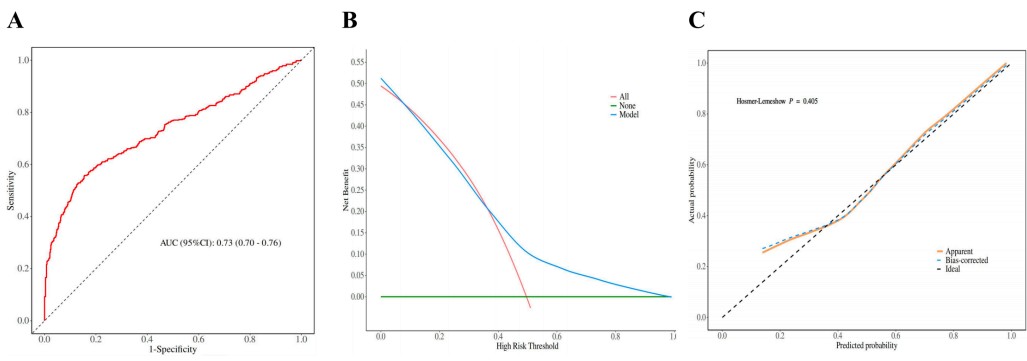

**Figure 2  Model discrimination and performance in the validation set.**

maximum net benefit is 0.52; the calibration curve established the model's accuracy and dependability (Figs. 2B and 2C).

## DISCUSSION

This study assessed the connection between prostate cancer risk and NHHR. After controlling for all confounding variables, the results show that NHHR is still a substantial independent risk factor for prostate cancer. The same pattern was seen when we separated NHHR into four quartiles. Additionally, we discovered a nonlinear correlation between the two, with an inflection point of 1.824. The chance of developing prostate cancer dropped as the NHHR level rose. The change rate of prostate cancer risk dramatically dropped when the NHHR was higher than 1.824. Sensitivity analysis demonstrated how reliable the study's findings were.

Lipid parameters have been linked to cancer in an increasing number of research (*Suh et al., 2023*; *Zhang et al., 2023*), however, the effects of lipids on cancer are complex. On the one hand, changes in lipid composition can change membrane fluidity and regulate cell growth rate (*Currie et al., 2013*), on the other hand, the communication between cells and tumor microenvironments can be enhanced by bioactive molecules (*Deep & Schlaepfer, 2016*). Prostate cancer cells overcome various barriers in the metastatic cascade by modifying lipid metabolism (*Cancel et al., 2022*; *Zhang et al., 2023*). NHHR is a new composite lipid parameter, which reflects the relative ratio of NHDL-C and HDL-C. Among them, NHDL-C contains cholesterol in all atherogenic lipoprotein particles, such as low density lipoprotein cholesterol, very low density lipoprotein cholesterol and chylomicrons (*Contois, Warnick & Sniderman, 2011*). Therefore, NHHR is better able to reflect the lipoprotein level of patients (*Sniderman et al., 2010*; *Su, Kong & Peng, 2019*). Studies have shown that NHHR is associated with the risk of many diseases such as diabetic nephropathy, depression (*Qi et al., 2024*), and breast cancer (*Zhang et al., 2024* and *Luo et al., 2024*). Likewise, we discovered a nonlinear dose–response association between the risk of prostate cancer and NHHR level in our retrospective analysis. The risk of prostate cancer decreased as the NHHR level rose. The influence on the risk of prostate cancer progressively diminished when the NHHR level surpassed the 1.824 threshold.

Furthermore, further ROC curve research revealed that NHHR had a remarkable value in predicting the risk of prostate cancer, the decision curve demonstrated that NHHR had a high clinical application potential, and the calibration curve confirmed that the model was accurate and reliable. Although lipids and cancer risk have been extensively investigated, it remains uncertain whether NHHR can be utilized as a marker to predict prostate cancer risk. To some extent, our study fills a void left by prior research. Our findings may have significant therapeutic implications for prostate cancer prevention and treatment. When compared to standard lipid metrics, NHHR can more fully reflect lipid levels and is more readily available. Only total cholesterol and HDL-C levels should be obtained. These signs can be discovered by regular blood tests. However, the underlying mechanism of NHHR impacting prostate cancer is complex, and future research will need to be more complete to demonstrate this process.

### Strengths and limitations

The current study has several strengths. Firstly, we eliminated potential confounders using propensity score matching, ensuring data comparability. Furthermore, we made significant model tweaks to assure the results' stability, analyzing NHHR as a continuous variable and quartiles, and the results remained robust, with more subtle variations seen. We further validated the results with modified Poisson regression. Finally, we evaluated NHHR's predictive value and clinical utility using ROC curves, decision curves, and calibration curves. By exposing the link between NHHR and prostate cancer, we found a more simple and useful prostate cancer biomarker. Based on the NHHR level, physicians can offer more realistic dietary recommendations for patients, as well as aid in the diagnosis and treatment of prostate cancer. The study has the following limitations: the data were collected from medical institutes in Xinjiang, China, and the results may solely reflect the relationship between lipid levels and prostate cancer risk in this region. Furthermore, there may be inconsistencies in the data collected from laboratory tests, and although though this study adjusted for all confounders, certain unavoidable confounders may still influence the results. Finally, because this is a retrospective study, the causal relationship between NHHR and the risk of prostate cancer cannot be established, and other essential variables, such as BMI and medication usage, cannot be obtained, potentially biasing the results.

## CONCLUSIONS

This study reveals a non-linear dose–response association between NHHR and prostate cancer risk, which could be linked to a lower risk of prostate cancer. This finding could be useful in the prevention of prostate cancer and the identification of high-risk people.

## ACKNOWLEDGEMENTS

The authors thank all patients and investigators.

### Funding

This work was supported by Xinjiang Uygur Autonomous Region Natural Science Outstanding Youth Program (Number: 2023D01E05), National Science and Nature Fund (Number: 82360476), the Key Project of Natural Science Foundation of Xinjiang Uygur Autonomous Region (Number: 2022D01D39), Xinjiang Uygur Autonomous Region Tianshan Talent Youth Top-notch Project (Number: 2022TSYCCX0026). The funders had no role in study design, data collection and analysis, decision to publish, or preparation of the manuscript.

### Grant Disclosures

The following grant information was disclosed by the authors:
Xinjiang Uygur Autonomous Region Natural Science Outstanding Youth Program: 2023D01E05.
National Science and Nature Fund: 82360476.
Key Project of Natural Science Foundation of Xinjiang Uygur Autonomous Region: 2022D01D39.
Xinjiang Uygur Autonomous Region Tianshan Talent Youth Top-notch Project: 2022TSYCCX0026.

### Competing Interests

The authors declare there are no competing interests.

### Author Contributions

- Ke Bu conceived and designed the experiments, performed the experiments, analyzed the data, prepared figures and/or tables, and approved the final draft.
- Jinru Wang conceived and designed the experiments, performed the experiments, prepared figures and/or tables, and approved the final draft.
- Xiaojie Zheng conceived and designed the experiments, performed the experiments, authored or reviewed drafts of the article, and approved the final draft.
- Kaige Zhang conceived and designed the experiments, performed the experiments, analyzed the data, authored or reviewed drafts of the article, and approved the final draft.
- Duolao Wang conceived and designed the experiments, performed the experiments, prepared figures and/or tables, and approved the final draft.
- Hengqing An conceived and designed the experiments, analyzed the data, authored or reviewed drafts of the article, and approved the final draft.
- Ning Tao conceived and designed the experiments, analyzed the data, prepared figures and/or tables, authored or reviewed drafts of the article, and approved the final draft.

### Human Ethics

The following information was supplied relating to ethical approvals (*i.e.*, approving body and any reference numbers):

The Ethics Committee of the First Affiliated Hospital of Xinjiang Medical University (20220308-166).

## Data Availability

The raw data is available in the Supplementary File.

## Supplemental Information

Supplemental information for this article can be found online at http://dx.doi.org/10.7717/peerj.19065#supplemental-information.

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
