# Peer review of "The association between non-high-density lipoprotein cholesterol to high-density lipoprotein cholesterol ratio (NHHR) and risk of prostate cancer: a retrospective study"

_PeerJ, doi:10.7717/peerj.19065_

## Round 0.1 · original submission · Major Revisions

Please address comments of all reviewers and provide your responses in a point wise manner.

·

Basic reporting

1. A more comprehensive and clearer overview of the literature will better place the study in context. This will greatly enrich the introduction and discussion section. For instance, in lines 58-62, authors can indicate which factors have been positively or negatively associated with prostate cancer. You can include systematic reviews and mention which areas of literature are still inconclusive.
2. Please report confidence intervals in lines 136-143.
3. Please report the finding of the 3rd and 4th quartile not having significantly lower odds of prostate cancer from the baseline in the text instead of just mentioning the 2nd quartile vs the baseline.
4. Table 3 was placed in the supplements instead of the manuscript. This is an important table and I suggest including it in the main text.
5. An AUC of 0.66 is not usually considered “high”. Please back this claim with some references.
6. Refining the language will help convey your message more clearly. Some examples where the language could be improved include lines 42, 72, etc. I suggested contacting a colleague who is proficient in English or a professional editing service. Similarly, please recheck the manuscript for typographical errors (e.g. line 50, 185-188, 286-287 etc.)
7. Please improve quality/resolution of images in Fig 2.
8. Line 188 has no citation (later studies have supported this view)
9. Line 190-191, this citation discusses T2DM but not cancer.
10. Please cite R packages used for analysis, if any.

Experimental design

1. Good that sensitivity analysis was done to confirm findings.
2. Can the authors explain why they used the covariates used (for example, differential WBC counts)? 3. Further, important covariates such as BMI and lipid-lowering therapy are missing from the regression models. Literature has suggested that lipid-lowering therapies have an effect on prostate cancer risk, so this is crucial to adjust for and could even be used to disqualify patients. If this info is not available, this should be stated in the limitations.
4. Please clarify the timepoints when blood samples were taken. Please clarify eligibility criteria for controls- were these all admitted patients who were also biopsied but showed no prostate malignancy on pathology (e.g. BPH)? Please report when the blood samples were taken (e.g. at least X hours after fasting? on admission prior to biopsy?).
5. Please be more specific with “analyzer” (line 87)
6. Were there any missing data? How were they dealt with?

Validity of the findings

1. Have the assumptions of logistic regression been met? Please report the checks done if so (collinearity, linearity etc). In the manuscript, one conclusion stated is that NHHR is an independent risk factor for prostate cancer. However, looking at figure 1 shows us a complex non-linear relationship between NHHR and prostate cancer risk- higher odds of prostate cancer in the mid-range of NHHR values compared to both low and high NHHR. The results of logistic regression do not align with the more complex pattern uncovered by RCS analysis.
2. Please temper the conclusions section in the abstract considerably. This study is a retrospective cohort study, and does not establish risk and causation. I suggest also including/discussing the finding of a non-linear relationship with RCS and that the OR appears to be <1 before and >1 after the inflection point.
3. Given literature linking LDL and triglycerides with prostate cancer, it may be informative to compare the OR of using these markers vs the NHHR and see whether the NHHR provides additional discriminatory ability (https://pubmed.ncbi.nlm.nih.gov/33724557/, https://www.frontiersin.org/journals/oncology/articles/10.3389/fonc.2023.1113226/full#B1)). This is just a suggestion to improve the manuscript.
4. What is the variable "danhongsu" in the excel sheet of the data?
5. It may be worth mentioning that the OR for matched patients is quite close to 1 (0.84-0.99, p 0.049) and the p value is close to being nonsignificant.

·

Basic reporting

- In general, the written language is of acceptable quality, the article structure from introduction to conclusion is reasonable, and relevant references are provided. Figures and supplements are well-connected to the text, and data sharing is aligned with the journal.

- However, especially in the discussion section, there are redundancies and not informative stand-alone statements such as: “Later case-control studies have supported this view.” on line 190, or “NHHR has good predictive value in studies related to chronic diseases and diseases such as cancer”. I suggest rewriting such sections with more concision in mind.

- Important final note on reporting: Although in many cases the authors have provided associational language correctly, some missed cases wrongfully assign causal language in this observational study, such as line 172: “elevated levels of NHHR reduce the risk of prostate cancer”.

Experimental design

- The study’s research question is an effort to provide additional evidence on top of what is currently known about the relationship between circulating lipoproteins and prostate cancer and is within the scope of this journal’s aims. The methods are in general well-chosen for the study type and are reproducibly stated and explained.

- The authors do NOT provide references or justification for their choice of confounders. For instance, why does RBC count and not Hb/Hct? Is the reasoning clinical or related to data availability? I suggest providing well-regarded references for the observable confounders list and stating reasons for including and excluding specific variables.

- Provision of AUCs, calibration curves, and decision curve analysis are welcome. However, the cross-sectional nature of this study and lack of temporality between exposure and outcomes make diagnostic metrics more suited for evaluation. Therefore, the provision of sensitivity, specificity (or likelihood ratios), PPV, and NPV will be more informative to the audience.

Validity of the findings

- Although, as stated before, the work could benefit from more concision in the discussion and remedy of some causal language, on the whole, the discussion puts the results within the context of previous research properly.

- A limiting factor in the study findings' generalizability is the lack of adequate attention to two major variables that necessitate extra analysis within the sensitivity analysis framework. A) Presence of hyperlipidemia and/or receiving statins or other lipid-lowering drugs; B) history of other cancers. I suggest the authors repeat the analysis of both subgroups within each variable (positive and negative for the covariates) and assess potential interactions.

- In case providing such information is not possible, the authors should clearly state the reasons and implications in the limitation section of the paper.

Additional comments

This study or any of its references do not establish whether the non-HDL to HDL ratio is the most relevant lipid-related variable for this association. As cardiovascular studies have shown us, there is reason to believe non-HDL cholesterol values alone are informative enough as they already result from the subtraction of HDL from total cholesterol values. Therefore, the use of ratio instead of direct values of this lipoprotein may not be necessary. Also, direct lipid values (in their original or log-transformed form) may be better-behaved and linear which will obviate the need for the use of a more complicated ratio.

I suggest if feasible, the authors repeat the analysis using non-HD lipoprotein values and compare the results with the ratio. It has a chance to provide valuable insights.

Reviewer 3 ·

Basic reporting

The study on the association between non-high-density lipoprotein cholesterol to high-density lipoprotein cholesterol ratio (NHHR) and prostate cancer risk offers some novel insights but also presents several weaknesses in its design, analysis, and conclusions.

o Retrospective studies are inherently prone to bias and confounding, which even propensity score matching cannot fully eliminate.
o The lack of prospective data limits causal inference.
o The study is limited to a single geographic region (Xinjiang, China), making the findings less generalizable to other populations, especially those with different dietary and genetic predispositions.
o Although the study claims extensive adjustment for confounders, significant risk factors for prostate cancer (e.g., dietary habits, family history, androgen levels, and socioeconomic factors) were not explicitly accounted for.
o The study does not provide mechanistic data linking NHHR with prostate cancer, leaving the observed associations speculative.
o The ROC curve analysis yielded an AUC of 0.64, indicating poor discriminatory ability of NHHR for prostate cancer risk. An AUC below 0.7 is typically considered inadequate for clinical use.
o The restricted cubic spline analysis revealed a non-linear relationship, but the interpretation of the inflection point (1.824) is unclear. The biological significance of this threshold is speculative and unsupported by mechanistic insights.
o The reported p-values (e.g., 0.049) are very close to the threshold of significance, raising concerns about the robustness of the results, especially given the potential for type I errors in large datasets.
o The quartile analysis shows a protective association only in the second quartile, which is not consistently observed across all quartiles, making the dose-response relationship less convincing.
o The conclusion that controlling lipid levels could reduce prostate cancer risk is unsupported due to the absence of intervention studies or longitudinal data.
o The modest AUC undermines the claim that NHHR has "good predictive value" for prostate cancer risk.
o While some limitations (e.g., regional data) are acknowledged, the broader implications of weak predictive value and potential residual confounding are underemphasized.

While NHHR as a lipid parameter has been studied in the context of cardiovascular diseases and metabolic syndrome, its application in cancer risk prediction is novel. However, the study does not demonstrate a substantial improvement over existing prostate cancer biomarkers such as PSA (prostate-specific antigen) in predictive performance. Previous studies linking lipids and prostate cancer often explore more established parameters like triglycerides, LDL-C, or HDL-C, suggesting NHHR’s incremental utility requires further justification. Additionally, the study's weak predictive value, methodological shortcomings, and absence of mechanistic insights significantly limit the reliability and applicability of its findings.

Experimental design

Please see the detailed reporting

Validity of the findings

no comment

---

## Round 0.2 · Minor Revisions

Please incorporate supplementary material as suggested by the reviewer 2.

·

Basic reporting

The modifications made following the first round of reviewers’ comments are satisfactory and will improve reporting.

Experimental design

- The explanations provided for variable choices are satisfactory and should be incorporated into the method section of the manuscript.
- The diagnostic table provided is very informative. I suggest, upon editors’ approval, the authors include it as a supplementary material.

Validity of the findings

- Like the previous points, I find the explanations on the choice of variables and patient population satisfactory. They need to be incorporated in the text, in the method section, and/or limitations.

Additional comments

While I admire the authors' additional analysis of non-HDL cholesterol, it is unclear whether the adjusted model included LDL as a covariate (like the primary model in the manuscript text). I suggest repeating the analysis without any cholesterol-related variable other than non-HDL cholesterol. Regardless of the results, this will be an essential side analysis to report to the audience (preferably as a supplement).

Reviewer 3 ·

Basic reporting

Authors have revised the manuscript appropriately

Experimental design

no comment'

Validity of the findings

no comment'

Additional comments

no comment'

---

## Round 0.3 · accepted · Accept

The authors have addressed all of the reviewers' comments and manuscript is ready for publication.